



# Technical Note: Emission factors, chemical composition and morphology of particles emitted from Euro 5 diesel and gasoline light duty vehicles during transient cycles

Evangelia Kostenidou[1], Alvaro Martinez-Valiente[2], Badr R'Mili[1], Baptiste Marques[1], Brice Temime-Roussel[1], Michel André[3], Yao Liu[3], Cédric Louis[3], Boris Vansevenant[3], Daniel Ferry[4], Carine Laffon[4], Philippe Parent[4] and Barbara D'Anna[1]

[1] Aix-Marseille Université, UMR 7673 CNRS, LCE, Marseille, France
[2] IRCELYON, UMR 5256 CNRS, Université de Lyon, Villeurbanne, France
[3] AME-EASE, Univ Gustave Eiffel, IFSTTAR, Univ Lyon, F-69675 Lyon, France
[4] Aix-Marseille Université, CNRS, CINaM, Marseille, France

*Correspondence to*: Evangelia Kostenidou (evangelia.kostenidou@univ-amu.fr) and Barbara D'Anna (barbara.danna@univ-amu.fr)

**Abstract.** Changes in engine technologies and after-treatment devices can profoundly alter the chemical composition of the
emitted pollutants. To investigate these effects, we characterized the chemical composition of particles emitted from three diesel and four gasoline Euro 5 light duty vehicles on a chassis dynamometer facility. Black carbon (BC) was the dominant emitted species with emission factors (EFs) varying from 0.2 to 7.1 mg km$^{-1}$ for gasoline cars and 0.003 to 0.08 mg km$^{-1}$ for diesel cars. For gasoline cars, the organic matter (OM) EFs varied from 5 to 103 µg km$^{-1}$ for direct injection (GDI) vehicles, and from 1 to 8 µg km$^{-1}$ for port fuel injection (PFI) vehicles, while for the diesel cars it ranged between 0.15 and 65 µg km$^{-1}$.
Cold-start cycles and more specifically the first minutes of the cycle, contributed the largest fraction of the PM including BC, OM and Polycyclic Aromatic Hydrocarbons (PAHs). More than 40 PAHs, including methylated, nitro, oxygenated and amino PAHs were identified and quantified in both diesel and gasoline exhaust particles using an Aerodyne High Resolution Time-of-Flight Aerosol Mass Spectrometry (HR-ToF-AMS). The PAHs emissions from the GDI technology were a factor of 4 higher compared to the vehicles equipped with a PFI system during the cold start cycle, while the nitro-PAHs fraction was
much more appreciable in the GDI emissions. For two of the three diesel vehicles the PAHs emissions were close to the detection limit, but for one, which presented an after-treatment device failure, the average PAHs EF was 2.04 µg km$^{-1}$. Emissions of nanoparticles (below 30nm), mainly composed by ammonium bisulfate, were measured during the passive regeneration of the catalyzed diesel particulate filter (CDPF) vehicle. TEM images confirmed the presence of ubiquitous nanometric metal inclusions into soot particles emitted from the diesel vehicle equipped with a fuel borne catalyst - diesel
particulate filter (FBC-DPF). XPS analysis of the particles emitted by the PFI car revealed both the presence of heavy elements (Ti, Zn, Ca, Si, P, Cl), and disordered soot surface with a significant concentration of carbon radical defects having possible consequences on both chemical reactivity and particle toxicity. Our findings show that different after-treatment technologies



have an important effect on the level and the chemical composition of the emitted particles. In addition, this research highlights the importance of the particle filter devices condition and their regular checking.

## 1 Introduction

On-road diesel and gasoline vehicles are an important source of urban air pollution, releasing fine particulate matter (PM1) and gaseous pollutants into the atmosphere (Dallmann and Harley, 2010; Borbon et al., 2013; Platt et al., 2014; Argyropoulos et al., 2016; Hoffman et al., 2016; Gentner et al., 2017). Light-duty vehicle pollutants have been associated with adverse effects on human health inducing cardiovascular, respiratory and also cognitive diseases (Hime et al., 2018 and references therein). At the same time, modern vehicles produce also $CO_2$ and BC which impact the climate (Lelieveld et al., 2019). Vehicle emissions received a lot of attention in the last years and diverse approaches have been used for their quantification, including tunnel studies (Grieshop et al., 2006; Lawrence et al., 2013; Smit et al., 2017), remote sensing or roadside measurements (Jimenez et al., 2000; Peitzmeier et al., 2017; Ropkins et al., 2017), on road (chase) measurements (Canagaratna et al., 2004; Morawska et al., 2007; Hudda et al., 2013; Karjalainen et al., 2014), on board measurements (Huo et al., 2012; Chikhi et al., 2014) and chassis dynamometer facilities (e.g., Andersson et al., 2014; Collier et al., 2015; Karjalainen et al., 2014; Saliba et al., 2017; Jaworski et al., 2018; R'Mili et al., 2018).

Improved information about the chemical composition of PM is essential to understand the source contributions, to implement mitigation measures and to assess health protection programs. PM vehicle emissions are mainly composed of BC and organic aerosol (OA) due to the incomplete combustion of fuel and lubricating oil. Less abundant components of vehicles exhausts include sulfate and metal in traces (Maricq, 2007; Cheung et al., 2010). Sulfur and trace elements such as Fe, Zn, P, Mg and Ca are commonly used as additives to lubricant oils (Maricq, 2007; Rönkkö et al., 2014). The above elements have been correlated to the oxidative potential indicating the toxic nature of these emissions (Cheung et al., 2010).

PAHs have been measured in both modern diesel and gasoline engine exhaust (Zielinska et al., 2004; Cheung et al., 2010; Alkourdi et al., 2013; Huang et al., 2013; Herring et al., 2015; Muñoz et al., 2018) and have been recognized as carcinogenic for humans (IARC, 2010). Exposure to PAHs is associated with excess risk of lung cancer as well as other adverse health effects including bronchitis, asthma, heart disease and reproductive toxicity (IARC, 2012, 2013; 2014). More recently oxygenated (OPAHs) and nitro- (NPAHs) polycyclic aromatic hydrocarbons have received increasing attention because of their cytotoxicity, immunotoxicity, carcinogenicity and mutagenicity (Durant et al., 1996; IARC, 2012; 2013; 2014). In general, gasoline and diesel PM, and more specifically the ultrafine particle fraction (below 100 nm) has been associated to an increased toxicity given its higher surface area and its greater ability to adsorb organic chemicals and metals (Mills et al., 2011; Cassee et al., 2013; Chen et al., 2016; Tyler et al., 2016).

Since 2012 the European market share of diesel light-duty vehicles started to decrease, dropping down to 35.9% of new vehicles sales in 2018, while the demanding of gasoline cars increased to 56.7% (ACEA 2019/2020). This trend continues despite the efficient PM reduction achieved by diesel cars, due to the combination of diesel oxidation catalysts (DOCs) and



diesel particulate filter (DPFs) (Gordon et al., 2013). Market share of gasoline vehicles equipped with direct injection (DI) system is steadily increasing due to a better fuel efficiency and consequently lower CO2 emissions than in conventional port fuel injection (PFI) engines (Alkidas, 2007; Myung et al., 2012; Liang et al., 2013). In GDI engines, the fuel is injected at higher pressures and it is mixed less uniformly with the incoming air. As a consequence of this inhomogeneous combustion, these vehicles release a large number of soot nanoparticles with diameters below 100 nm  (Karjalainen et al., 2014; Zhu et al.,

2016; Zimmerman et al., 2016; Platt et al., 2017; Saliba et al., 2017; Du et al., 2018). Gasoline vehicles do also emit NH3 generated by the three-way catalyst (TWC) (e.g., Heeb et al., 2006; Suarez-Bertoa et al., 2017) which might highly enhance the new particle formation in the environment. Despite all the previous studies for the characterization of the vehicles' emissions, an integrated and comprehensive analysis for the understanding of these emissions is missing.

The present study provides a detailed description and comparison of PM emissions from three diesel and four gasoline (both

GDI and PFI) Euro 5 light-duty vehicles tested on a roll bench chassis dynamometer facility. The chemical speciation of particulate matter emitted from the seven vehicles (BC, organics, sulfate, ammonium, nitrate and PAHs) was accomplished using real time and high-resolution instrumentation. We explored the emission profile of each species and we evaluated the corresponding EFs in terms of main chemical species including BC, PAHs and OA. In addition, we investigated the morphology, the surface structure and the elemental composition of PM through offline analysis.

**2 Experimental**

**2.1 Vehicles and infrastructure**

Four gasoline and three diesel Euro 5 light duty vehicles were investigated; their names and specifications are listed in Table 1. The passenger vehicles were either rent from a local rental car or they were privately owned, and their mileage ranged from 9500 to 103000 Km. We tested three GDI cars and one PFI vehicle. All diesel cars were equipped with a DOC and a DFP; two

were catalyzed (CDPF) and one was fuel borne catalyst (FBC-DPF). Commercial fuel was purchased from the same gas station to minimize the influence of fuel composition on emissions. The fuels and the lubricant oils were analyzed by Inductively Coupled Plasma Mass Spectrometry (ICP-MS) analysis for the measurement of their elemental composition.

The experiments were carried out on the roll bench chassis dynamometer owned by the Environment, Planning, Safety and Eco-design Laboratory (EASE) of the French Institute of Science and Technology for Transport, Development and Networks

(IFSTTAR).  Three types of dilution systems were used a) a Constant Volume Sampler (CVS) and two VLK 10 Palas for vehicles D1 and GDI1, b) a Fine Particle Sampler (FPS 4000, Dekati Ltd) with two dilution stages for vehicles D3 and GDI3 and c) an ejector dilution of one stage (with hot air) for vehicles D4, PFI4 and GDI5. The dilution ratios for each system were calculated based on CO2 concentrations and are provided in Table 1.

The emissions of vehicles D3 and GDI3 were tested using the Worldwide Harmonized Light Vehicles Cycle (WLTC), while

the vehicles D1, D4, GDI1, GDI5 and PFI4 were evaluated using the Common Artemis Driving Cycles (CADC hereafter Artemis) (André, 2004) which allows to evaluate separately the contributions of urban, rural and motorway driving conditions.



A suite of instrumentation was deployed for the characterization of both particle and gas phase emissions. A stainless-steel sampling line of 10 mm inner diameter and 5-6 m long was used. The line was heated at 80-120o C, while the total flow was approximately 60 l min$^{-1}$. Before reaching the instruments, the line was split in two parts: one for the PM measurements, which was kept at room temperature, and a second one for the gas phase measurements kept at 80-120o C. In this work, we focus on the particle phase only.

## 2.2 Instrumentation

Four experimental campaigns were conducted using different combination of measurement systems. Thus, the instrumentation configuration was not the same for all the vehicles tested Table 1).

### 2.2.1 HR-ToF-AMS and c-ToF-AMS

An Aerodyne High Resolution Time-of-Flight Aerosol Mass Spectrometer (HR-ToF-AMS) (DeCarlo et al., 2006) or a compact Time-of-Flight Aerosol Mass Spectrometer (c-ToF-AMS) (Drewnick et al., 2005) were deployed for the size-resolved chemical composition of the non-refractory species (e.g., organics, sulfate, ammonium, nitrate). The vaporizer temperature was set at 600-650oC and the tungsten filament for electron ionization was run at an accelerating voltage of 70 eV. The sampling time resolution was set between 20 and 40 sec in V-mode.

### 2.2.2 Aethalometer and MAAP

An Aethalometer AE 33-7 (Magee scientific) or a Multi-Angle Absorption Photometer (MAAP 5012, Thermo Fisher Scientific) were used for black carbon measurements at a wavelength of 670 and 880 nm respectively. The time resolution for the Aethalometer was 1 sec, while for the MAAP was 2 min. The MAAP data, characterized by high BC mass concentrations, were corrected using the method proposed by Hyvärinen et al. (2013).

### 2.2.3 Exhaust gas sampling and analysis devices

A mini-particle sampler (MPS) was used to collect exhaust particles. This technique, based on filtration through TEM-porous grids (Holey carbon films and Quantifoil, AGAR Scientific), enables sampling of particles directly on a specific support minimizing additional sample preparation procedure and sampling artifacts (R'Mili et al., 2013). The sampling was performed at specific times of the cycle for example at the first minutes of the cold start or during a few minutes of the motorway with a flowrate of 0.3 l min$^{-1}$. The deposited particles were then investigated by Transmission Electron Microscopy (TEM), coupled with energy-dispersive X-ray (EDX). Grids from the D1 and GDI1 vehicles were analysed using a JEOL 2010F microscope operated at 200 kV, while all the other samples were analysed using a JEOL JEM2010 microscope, fitted with a LaB6 electron gun under a 200 kV accelerating voltage and an edge-to-edge resolution of 0.23 nm at 200 kV. The instrumentation included also an EDX spectrometer (BRUKER, Quantax) with a XFlash® silicon drift detector, which has a resolution of 65 eV and 73 eV for the Kα transition of carbon and fluoride respectively.





Particles emitted by the PIF4 vehicle were further analysed by XPS recorded under ultra-high vacuum using a Resolve 120 hemispherical electron analyser (PSP Vacuum) and an un-monochromatized X-ray source (Mg Kα at 1253.6 eV, PSP Vacuum) operated at 100 W at an incidence angle of 30° with respect to the analyser axis. This X-ray excitation energy and detection

geometry correspond to an analyzed depth of about 1 nm at the C1s and O1s lines. Survey spectra were collected at a pass energy of 50 eV and an energy step of 0.2 eV, while the other lines were collected at 20 eV pass energy and a step of 0.1 eV. The XPS lines were deconvoluted with the CasaXPS program, after Shirley-type or linear background subtraction. Quantitative estimations of the samples composition were done after correction by the relative sensitivity factors (RSF) provided in the program.

**2.3 Instrumentation**

**2.3.1 Organics and PAHs**

The AMS data were analyzed with SQUIRREL v1.60A and PIKA v1.20A with Igor Pro 6.37 (Wave-Metrics). For the organic species, we used the fragmentation table of Aiken et al. (2009). For the vehicles D1, D3 and GDI3 the mass spectra are provided in unit mass resolution (UMR), while for the vehicles D4, PFI4 and GDI5 the mass spectra are given in high resolution (HR).

Since the majority of the signal at m/z 44 is practically due to the gaseous $CO2$ the signal at m/z 44 was removed from the fragmentation table. In addition, we removed the m/z's 73, 147, 207, 221 and 281 as they were related to polydimethylsiloxane ($SiO(CH3)2$) contaminations due to conductive silicone material present in parts of the tailpipe (Timko et al., 2009; 2014). For the quantification of the aerosol and the PAHs mass concentration, a collection efficiency (CE) of 1 was used following previous engine exhaust studies (Canagaratna et al., 2004; Slowik et al., 2004; Dallmann et al., 2014; Eriksson et al., 2014;

Bruns et al., 2015; Herring et al., 2015). The relative ionization efficiency (RIE) of the PAHs was set to 1.4, as measurements of four PAHs resulted in an RIE between 1.35 and 2.1 (Slowik et al., 2004; Dzepina et al., 2007). PAHs analysis was carried out using Herring et al. (2015) methodology for vehicles D4, PFI4 and GDI5 (HR-ToF-AMS data). Briefly, each PAH compound concentration ($C_i$) can be estimated based on the relative abundance ($f_{A,i}$) of the PAH molecular ion ($C_{ion,i}$) in the reference mass spectra (measured for single PAH compounds) by Eq (1):


$$C_i = \frac{C_{ion,i}}{f_{A,i}} \tag{1}$$

Even though for most of the PAHs the $f_{A,i}$ is unknown, some laboratory-measured PAH spectra using the AMS do exist (Alfarra, 2004; Aiken et al., 2007; Dzepina et al., 2007). These spectra are for pyrene ($C_{16}H_{10}$, *m/z* 202), fluoranthene ($C_{16}H_{10}$, *m/z* 202),

2,3-benzofluorene ($C_{17}H_{12}$, *m/z* 216), 1-methylpyrene ($C_{17}H_{12}$, *m/z* 216), triphenylene ($C_{18}H_{12}$, *m/z* 228), 10-methylbenz[a]anthracene ($C_{19}H_{14}$, *m/z* 242), benzo[e]pyrene($C_{20}H_{12}$, *m/z* 252), and benzo[ghi]perylene ($C_{22}H_{12}$, *m/z* 276), and their relative abundance ($f_{A,i}$) is summarized in Herring et al. (2015). In this work we used the average relative abundance of the above measured compounds, which corresponds to an $f_{A,i}$ of 26.5%. Using this assumption we introduce an error of ±40%





(taking account the minimum and the maximum $f_{A,i}$ reported for the AMS measured PAH spectra). An example of the HR
mass spectra fitting for naphthalene, methyl-naphthalene, anthracene and nitro-anthracene is given in Figure S1.

**2.3.2 Emission factors**

The EF of each species during the cycle was calculated using the Eq. (2):

$$EF = \frac{DR}{D} \int C(t) * Q_{ex}(t) * dt \qquad (2)$$

were C(t) is the mass concentration of the pollutant, $Q_{ex}(t)$ is the exhaust flow rate at the tail pipe measured by the CVS or the
FPS, DR is the external dilution before the entrance of the instrumentation, and D is the distance of the cycle (4.51 km for
Artemis urban cycle, 23.8 km for Artemis motorway cycle and 23.25 km for WLTC).

**3 Results**

**3.1 AMS chemical composition**

**3.1.1 Time series profiles**

Figure 1 shows the particle mass concentration transient profile of the BC, organics, sulfate, nitrate and ammonium measured
by the HR-ToF-AMS for the GDI5 vehicle during Artemis cold urban, hot urban and motorway cycles. The mass
concentrations have been corrected for the dilution in front of the instrumentation. The particle phase emissions were mainly
composed of BC (96.8-98%), while the organic fraction accounted for 1.9-3.1%; ammonium, sulfate and nitrate were
approximately 0.1%. The highest mass concentrations of BC and organics, 1600 µg m⁻³ and 120 mg m⁻³ respectively, were
observed during the first 1-2 minutes of the cycle, due to the cold engine and thus low catalyst efficiency, which is in agreement
with previous studies (e.g., Weilenmann et al., 2009; Clairotte et al., 2013; Collier et al., 2015; Karjalainen et al., 2016; Louis
et al., 2016; Pieber et al., 2018). GDI3 particulate mass concentrations in the exhaust flow were measured during WLTC
(Figure S2): BC contributed 83-98 % to the total PM mass, while the organic fraction ranged from 1.8 to 14% of the PM. The
remaining fraction 0.2-3% was composed of ammonium, sulfate and nitrate. The emitted PM concentrations were comparable
to the values measured for the GDI5 vehicle during Artemis/Cold Urban.
PM emissions from the PFI4 vehicle are shown in Figure S3. The organic and nitrate mass concentrations were a factor of 10
lower in comparison to those for GDI5 vehicle. High GDI PM emissions have been also reported in previous studies (e.g.,
Zhu et al., 2016; Saliba et al., 2017; Du et al., 2018), and were explained by the incomplete volatilization and mixing of the
fuel in the combustion chamber (Fu et al., 2014; Chen et al., 2017; Saliba et al., 2017).
Figure 2 shows the PM emissions of the D1 car (equipped with a CDPF) in terms of (a) chemical composition and (b and c)
particle size distribution during a cold urban and three consecutive motorway cycles. The cold cycle was characterized by
relatively high BC and organic matter emissions, reaching concentrations of 300 and 50 µg m⁻³, respectively. During the cold
urban cycle BC accounted for 94% of the total mass concentration and organics only for 4%, while during the motorway cycle





the contribution of BC decreased to 85% while ammonium bisulfate increased to (6%) and organics to (8%). The three

motorway cycles showed good repeatability, characterized by a first release of BC followed by emissions of ammonium

bisulfate and organics nanoparticles (15 nm mean diameter). This behavior was interpreted as a passive regeneration of the

DPF occurring at the high temperatures reached during the cycle. Similar observations have been reported during regeneration

of diesel cars equipped with CDPF (R'Mili et al., 2018).

The emission profiles during WLTC cycles from a second CDPF vehicle (D3) are shown in Figure S4. This vehicle was

characterized by very low emissions, demonstrating the efficiency of the after treatment devices. Emissions were observed

during few accelerations; the organic mass concentration remained always below 20 μg m$^{-3}$, while ammonium bisulfate

concentrations reached maximum values of 50 μg m$^{-3}$.

The emission of sulfate containing particles from the two CDPF vehicles was explained by the presence of the catalyst on the

DPF walls. It has been proposed that during acceleration or hot engine combustion periods sulfur can be released and converted

into SO3 by the catalyst, forming successively sulfuric acid and/or bisulfate/sulfate ammonium ultrafine particles  (e.g., Bikas

and Zervas, 2007; Bergmann et al., 2009; Arnold et al., 2012; R'Mili et al., 2018).

PM emissions from the D4 vehicle equipped with an FBC catalyst (Figure S5) were relatively high: OA, nitrates and sulfate

reached 300, 90 and 40 μg m$^{-3}$, respectively. The high PM concentrations were interpreted as a possible failure in the after-

treatment system and will be further discussed in section 3.3. This was supported by the relatively higher emissions of $CO_2$,

CO, $NO_x$ and THC in comparison to the rest diesel cars (Table S1).

### 3.1.2 Organic mass spectra

Figure 3 shows the HR-AMS mass spectra for the GDI5 and the D4 vehicle during the first and last 2 minutes of each cycle.

The mass spectra were  characterized by the ion fragments $C_nH^+_{2n+1}$ (m/z = 29, 43, 57, 71, 85…) typical of saturated alkyl

compounds (n-alkanes and branched alkanes), $C_nH^+_{2n-1}$ (m/z = 27, 41, 55, 69, 83, 97…) typical of unsaturated aliphatic

compounds (cycloalkanes, alkenes), and $C_nH^+_{2n-3}$ (m/z = 67, 81, 95, 109…) typical of bicycloalkanes and alkynes (McLafferty

and Turecek, 1993). These spectra are consistent with the signatures found in both gasoline and diesel exhaust emissions (e.g.,

Canagaratna et al., 2004; Mohr et al., 2009; Chirico et al., 2011; Platt et al., 2013; Collier et al., 2015; Dallmann et al., 2014;

R'Mili et al., 2018), and arise from both unburned fuel, lubricating oil, and their partially oxidized products (Maricq, 2007).

During some periods of the cycle, the m/z ratios of 43/41, 57/55 and 71/69 were relatively high with values of 1.50, 1.72 and

1.19 during the GDI5 cold start (Figure 3a), and 1.30, 1.32 and 0.85 during the D4 hot engine regimes (Figure 3b). Comparing

our mass spectra with pure gasoline, diesel and lubricant oil mass spectra analyzed with a similar instrument (R'Mili et al.,

2018), and knowing that the fuels contain high concentrations of n-alkanes, while lubricating oils tend to contain mostly

cycloalkanes (Tobias et al., 2001; Isaacman et al., 2012) we concluded that both GDI5 and D4 emitted randomly oil droplets

(see also section 3.2 for TEM images).

Hydrocarbons ion fragments accounted for 77 to 90% of the OA fraction for the GDI5 (Figure 3a), for 83-88% for the D4

(Figure 3b) and for 56-87% for the PFI4 vehicles (Figure S6). Similar mass spectra were observed for the GDI3 and D1 vehicle




(Figure S7 and Figure S8, UMR mass spectrum). The OA concentration emitted from the D3 car was very low and the high uncertainty was associated to the corresponding AMS mass spectrum.

For all of the gasoline cars, sulfur containing organic fragments at $m/z$ 45 (CHS$^+$, 44.979), 46 (CH$_2$S$^+$, 45.987) and 47 (CH$_3$S$^+$, 46.995) were detected. They accounted for approximately 2-4% of the organic mass fraction for the GDI5 (Figure 3a) and 6-7% to the organic mass for the PFI4 (Figure S7). For the GDI3 car (Figure S5), a high $m/z$ 45 contribution was detected at the beginning of the hot start WLTC, but the spectrum was acquired with a c-ToF-AMS and therefore the signal can be assigned to both oxygenated (CHO$_2$$^+$ and C$_2$H$_5$O$^+$) and organosulfur (CHS$^+$) fragments. Sulfur containing ion fragments were mostly emitted from hot engines (end of urban cycle and motorway cycle) and are tentatively explained by the release of some lubricant oil.

Table 2 presents the correlations between the mass spectra of the tested vehicles with those of previous studies (AMS mass spectra database). A very good correlation was found between the mass spectra from diesel and gasoline vehicles (Canagaratna et al., 2004; Mohr et al. 2009; R'Mili et al., 2018) and PMF factors related to fresh traffic emissions (Mohr et al., 2012; Kostenidou et al., 2015; Kaltsonoudis et al., 2017). The R$^2$ ranged between 0.72 and 0.92 (Table 2) for all cases.

### 3.1.3 PAHs

In total, 45 PAHs were identified for the GDI5, PFI4 and D4 vehicles during Artemis cycles (Table S2). The mass concentrations of all the PAHs during the cold cycle were considerably higher for GDI5 and D4 than for PFI4, with values of 1.66, 2.21 and 0.47 µg m$^{-3}$, correspondingly. Slightly lower mass concentrations were observed during the hot cycles. For the D1 and D3 vehicles, the PAHs signal was close to the detection limit, demonstrating that after-treatments devices (DOC and DPF) efficiently reduce PAHs emission from light-duty diesel engines. The remarkable difference of the three diesel vehicles confirms that D4 indeed presented a failure in the after-treatment device. For the GDI3 vehicle, the identification of individual PAHs was not possible since the data were collected with a c-ToF-AMS (UMR mass spectra).

Figure 4 presents the relative contribution of five PAHs families for the GDI5, PFI4 and D4 cars: unsubstituted PAHs (UnSubPAHs), methylated PAHs (MPAHs), oxygenated PAHs (OPAHs), nitro-PAHs (NPAHs) and amino PAHs (APAHs). The UnSubPAHs represented the most abundant group, accounting for 52 to 66% of the total PAHs, followed by MPAHs (14-35%), then OPAHs (5-19%), NPAHs (1-11%) and finally APAHs (1-6%). Table S3 demonstrates the individual PAHs fractions during the cold- and hot-start cycles. For all three cars, naphthalene emissions dominated, contributing from 9.6 to 19.1% of the total PAHs, which is in agreement with previous studies (e.g., de Abrantes et al., 2004; Vouitsis et al., 2009; Huang et al., 2013; Alves et al., 2015; de Souza and Corrêa 2016; Muñoz et al., 2018). Among the 3-rings PAHs species, acenaphthylene (4.3-9.7%), anthracene and its isomer phenanthrene (4.1-15.9%) were the most abundant; concerning the 4-rings PAHs, the major contribution derived from pyrene and from the isomers fluoranthene and acephenanthrylene (1.3-13.9%), while among the 5-rings PAHs, benzo[a]pyrene and all its isomers (0.4-3.8%) and benzo[ghi]fluoranthene (1-3.3%) were the most significant compounds. Some heavier PAHs as indio[1,2,3-cd]pyrene its isomer benzo[ghi]perylene (0.4-6.6%) and coronene (0.06-5.3%) were mostly found in gasoline car emissions. Light PAHs have often been measured in exhaust



particles of light-duty vehicles (Ravindra et al. 2008; de Souza and Corrêa 2016; Muñoz et al., 2018), and their presence has been tentatively explained by incomplete fuel combustion (Lea-Langton et al., 2008; Ravindra et al., 2008) since these compounds are present in the fuel composition (Marr et al. 1999; de Souza and Corrêa 2016). During the gasoline hot cycles an increase of the 3- and 4- ring PAHs (anthracene, pyrene, paracylene and all its isomers) contribution was observed.

MPAHs accounted for 14 to 35% of the total PAHs, and were more abundant for the D4 vehicle; major contributions arose

from methyl- and dimethyl-naphthalene, methyl-phenanthrene, methyl-fluorene and ethyl-phenanthrene, which is in agreement with Muñoz et al. (2018). All these compounds have been recently associated to carcinogenic potency (Samburova et al., 2017). BaP and its isomers (Benzo[b]fluoranthene or Benzo[j]fluoranthene or Benzo[k]fluoranthene) contributed only 0.5 to 3.8% of the PAHs fraction of gasoline cars, while the signal was below the detection limit in the D4 emissions. Yet, recent studies suggest that BaP, as indicator compound, may highly underestimate the total carcinogenic potency of PAHs

mixtures (U.S. EPA, 2010; Samburova et al., 2017).

Figure 5 depicts  the transient profile of selected PAHs for the GDI5, PFI4 and D4 vehicles during Artemis cycles; these PAHs have been classified as carcinogenic or/and photomutagenic (compounds that cause mutagenicity after being exposed to visible or UV light) according to IARC 2010. Following BC and organics' emission trend, PAHs were also important in the first few minutes of the cold urban cycle and during acceleration periods of the motorway cycle or during fuel-rich combustion periods

in agreement with previous studies (Muñoz et al., 2018). The mass fractions of these carcinogenic and/or photomutagenic PAHs accounted for 27-49% for the GDI5, 29-30% for the PFI4 and 29-31% for the D4 vehicles.

A considerable fraction - up to 31% of the total PAHs - was functionalized and included OPAHs, NPAHs and APAHs. All technologies emitted an important fraction of OPAHs (up to 19%); anthraquinone was the most abundant in agreement with previous emission studies (Karavalakis et al., 2011) followed by fluorenone, indanone, dibenzofuran and dibenzopyran.

APAHs accounted for 1 to 6 % of the total PAHs fraction and were mostly emitted by gasoline cars. Major NPAHs were aminopyrene/carbazole and dibenzocarbazole/amino-benzopyrene, however very little is known about the car emissions of these compounds so far.

Nitro-anthracene and its isomer nitro-phenanthrene contributed up to 8% of the total PAHs in the GDI5 emissions, but only 1% in PFI4 and D4 vehicles. Nitro-fluorene, nitro-pyrene and nitro-chrysene were found in the car exhaust of all three vehicles,

and accounted for less than 1% of the total PAHs mass fraction. Even if present in small amounts, some of these compounds, as 6-nitrochrysene and 1-nitropyrene, are classified as possibly carcinogenic to humans (group 2B) (IARC, 2012; Bandowe and Meusel, 2017). Surprisingly, NPAHs, including nitro-pyrene, were considerably higher in GDI emissions than in those of diesel car, questioning the validity of using NPAHs such as 1-nitropyrene as markers of diesel emissions (Keyte et al., 2016).

**3.2 Off-line analysis: TEM and XPS**

Figure 6 presents TEM images of particles emitted by the different cars during cold cycles. Figures 6(a-c) and 6(d-f) show particles emitted from the GDI1 and GDI3 vehicles; the samples were collected during the first 120 seconds of the cycle and the dilution was around 40-46. Figure 6(g-i) shows three images of particles emitted from the D1 car; the corresponding sample





was collected for 300 seconds and the dilution was around 40. TEM images confirmed the quite higher emissions of soot particles (or BC) for the two GDI vehicles with respect to the diesel car, which is in agreement with BC emissions measured by the MAAP and the Aethalometer. As usually mentioned in the literature, soot particles are observed either as fractal branched chains and or as bigger agglomerates made of primary soot spheres of different sizes (Lapuerta et al., 2020). Primary soot particles with diameter of around 25 nm were observed for the gasoline cars during cold cycles (Figure 6a and b), while the diameter was significantly smaller (below 20 nm) during hot cycles (Figure S9c and S9f). The results are in a good agreement with previous literature, which reported primary soot particles with diameter in the range between 20-25 nm (Barone et al., 2012) and smaller sizes down to 16 nm (Mathis et al., 2004; Gaddam and Vander Wal et al., 2013) for gasoline exhaust particles. A slight decrease in the primary particles size with increasing temperature was observed, in agreement with recent studies (Cadrazco et al., 2019). It has been shown that the engine load has no effect on soot morphology (Lapuerta et al., 2020) as many other parameters may favor opposite trends and compensate each other. Indeed, a higher fuel-air ratio would tend to extend primary particles growth while a higher engine temperature would favor their oxidation and thus lead to smaller particle sizes (Ye et al., 2014). Similarly, increasing the injection timing leads to a decrease of primary particles size due to an increase of in-cylinder oxidation time (Xu et al., 2014). It is therefore difficult to unambiguously attribute the slight decrease in particle size observed only to the temperature effect.

Figure S9 (i) depicts soot particles from the D4 car; tiny sparse dark spots were ubiquitous within the soot particles and were interpreted as metal inclusions. Unfortunately, EDX could not reveal their chemical nature due to the very small amount of material in these inclusions as they were very small (typically less than 0.5 nm) and their spatial density was low. Nevertheless, we assume that these inclusions were metallic and resulted from the after-treatment device of the FBC-DPF vehicle (D4), which implies the use of additives made of metallic salts or organometallic compounds into the engine combustion chamber. Upon combustion, the additive produces nanoparticles of metal oxides that are mixed with soot particles and accumulated on the DPF walls. The role of these metals will be to reduce the DPF regeneration temperature (Ntziachristos et al., 2005; Majewski and Khair, 2006; Song et al., 2006).

Nearly spherical particles were observed for some of the cars: GDI3 (Figure 6b, Figure S9), D1 (Figure 6c) and D4 (Figure S9b). They were observed both during cold and warm cycles and they had variable sizes and shapes ranging from 100 nm to almost 1 μm. EDX analysis revealed that on average the droplets presented C and O as major components, followed by S which was enriched in few droplets. Minor components accounted also for calcium, phosphorus, sodium, silica. Only minor traces of zinc, iron, copper, chromium, aluminum and nickel were observed. Analysis of the lubricant oils for D1, D3 and GDI1, GDI3 are presented in table S4. Sulfur accounted around 0.12 and 0.14 wt% of the lubricant oil. Other components of the lubricant oil were calcium, phosphorus and zinc, and only traces of iron, silica and copper were found. The iron found in the used lubricant oil suggests erosion of the engine wear and exhaust line for both D1 and GDI1 vehicles. These findings are in line with previous studies that reported emissions of lubricant oil particles during transient driving conditions (Karjalainen et al. 2014; Rönkkö et al., 2014).





Particles emitted by the PFI4 car were analyzed by XPS. Figure S10 (a) shows the survey spectrum, from which the averaged elemental composition of the sample was derived (in wt.%). We noticed a strong C1s line at 285 eV (53.8 wt.%), a strong O1s line at 530 eV (23.9 wt.%), and weaker signals of Ti-2p doublet at 460 eV (8.3 wt.%), zinc Auger lines (265 eV) (3.6 wt %), calcium Ca2p (352 eV) (2.8 wt.%), silicon Si2p (110 eV) (2.5 wt.%), phosphorus P2p (140 eV) (1.5 wt.%). Other trace signals

(< 1 wt.%) of chlorine Cl2p (200 eV), nitrogen N1s (400 eV), silver Ag 3d (374 eV), and sulfur S 2p (168 eV) are also observed. From previous (unpublished) SEM-EDX analysis we know that Si is an artefact coming from the support plate while C, N, Ti, Cl, Ca, Ag, Zn, and a fraction of O originates from exhaust particles. Calcium, phosphorus, sulfur and zinc might derive from lubricant oil (Table S4), while Ti might originate from the washcoat of the catalytic converter. The weak N1s signal showed typical energy of amino groups confirming the presence of APAHs as observed from AMS chemical analysis

(Table S3).

Figure S10 (b and c) depicts the deconvolution of C1s (b) and O1s spectra (c). In the C1s spectrum the carbon speciation can be derived in terms of graphitic $sp^2$ carbon (at 284.5 eV), aliphatic $sp^3$ carbon (285.4 eV) and oxidized carbon in C-O-C bonds (ethers, alcohols; 286.4 eV), in C=O bonds (carbonyls, quinones, 287.5 eV), and acidic O=C*-OH bonds (288.9 eV) (Estrade-Szwarckopf, 2004). The analysis revealed a soot sample dominated by $sp^2$ hybridized carbon, the absence of the usual shakeup

line associated with graphitic structures, and a significant "defect" contribution (at 283.5 eV, 12% of the C1s signal) associated to carbon vacancies (Barinov et al., 2009), which indicates a significant concentration of carbon radical defects (Levi et al., 2015). All these elements hint to a structurally disordered soot surface, possibly having chemical toxicity or reactivity due to the presence of surface radicals. In addition, a rather high concentration of sp3 carbons (alkanes, 20 % of the total carbon) was detected at the surface of the particles, in agreement to what observed by AMS analysis (Figure S6). From the O1s spectrum

the relative contribution of the C=O carbonyl and carboxylic groups (532.1 eV), the C-O-C groups ethers and alcohols (533.2 eV), and the OH groups acids (534.3 eV) were derived. A strong contribution of Ti-O* in $TiO_2$ was detected at 530.2 eV coming from ashes. Oxidized calcium and silicon also contributed to the O1s spectrum as Si-O* and Ca-O* lines in the 533-535 eV range (Ni and Ratner, 2008; Yang et al., 2011). Using the C=O contribution at 532.2 eV - the only line not overlapped by the Si, Ca and Ti oxides- and the integrated intensity of the C1s line, we evaluated a soot surface oxidation by the ratio

O/(O+C), giving an oxidation rate of 10.8 %. This is in good agreement with Schuster et al. (2011) who found for Euro 4-5 soot particles oxidation rates between 5.5-11.5%.

### 3.3 Emission factors

Figure 7 shows the emission factors (EFs) for BC, organics, PAHs, sulfate, ammonium and nitrate for all the cars tested in this study. Table S5 summarizes these EFs in µg km$^{-1}$. Most of the particles were emitted during the cold start cycles (both Artemis

urban and WLTC) followed by motorway, hot WLTC, and hot urban cycle. Gasoline vehicles generally emitted higher BC concentrations compared to the diesel vehicles. The average BC EFs for the GDI5, GDI1 (during Artemis cold urban cycle) and the GDI3 (during WLTC cold) vehicles were 3.18, 7.14, and 5.7 mg km$^{-1}$, confirming a relative small variability among the vehicles having the same injection technology. These results are in a reasonably good agreement with previous studies,





such as Saliba et al. (2017) who reported elemental carbon (EC) EFs between 0.08 and 5.8 mg km$^{-1}$ for GDI light vehicles
(models 2012-2014) during a unified cycle cold start. Taking into account the cycle distance, the fuel consumption and the
fuel density (Table S6), the EFs were converted into mg kg$^{-1}_{fuel}$. The BC EFs during the cold cycles for GDI5, GDI1 and GDI3
were 51, 101 and 120 mg kg$^{-1}_{fuel}$, respectively. These values are in good agreement with Pieber et al. (2018) and Platt et al.
(2017), who reported BC EFs for GDI Euro5 light duty vehicles between 10 and 100 mg kg$^{-1}_{fuel}$ during cold start WLTC and
10-250 mg kg$^{-1}_{fuel}$ during cold start New European Driving Cycle (NEDC). The BC EF for the diesel D1 vehicle during Artemis
cold start cycle was 0.07 mg km$^{-1}$, while for the D3 was 0.01 mg km$^{-1}$ during a WLTC cold start in agreement with Platt et al.
(2017).

The OA EFs emitted by the GDI5, GDI3 and PFI4 gasoline cars were 66.3, 103.5 and 8.4 μg km$^{-1}$ respectively during the cold
start cycle. Converting these EFs into mass of fuel the corresponding EFs were 1.1, 2.2 and 0.2 mg kg$^{-1}_{fuel}$ for the GDI5, GDI3
and PFI4, respectively. This is in agreement with Pieber et al. (2018) who reported EFs between 1 and 10 mg kg$^{-1}_{fuel}$ for cold
start WLTC. The corresponding OA EFs of the D1, D3 and D4 vehicles were 11, 0.7 and 61 μg km$^{-1}$ (or 0.21, 0.02 and 0.94
mg kg$^{-1}_{fuel}$), correspondingly. The OA EF of the D3 was quite low and this is generally in agreement with Platt et al. (2017),
who found that the diesel vehicles equipped with a DPF emitted very low amounts of OA (less than 0.01 g km$^{-1}_{fuel}$). The D4
OA EF is close to the values of gasoline vehicles underlining the impact of DPF condition on PM emissions (see also below).

The PAHs EFs were measured for only 3 vehicles: GDI5, PFI4 and D4. The EFs for GDI5 were 1.5 and 1.1 μg km$^{-1}$ during
cold urban and motorway cycles correspondingly. The PFI4 vehicle emitted considerably lower PAHs, only 0.4 μg km$^{-1}$ for
cold urban and 0.04 μg km$^{-1}$ for motorway cycles. These values are in relatively good agreement with Cheung et al. (2010)
who reported PAHs EFs of 0.67 μg km$^{-1}$ from PFI gasoline cars. Alves et al. (2015) measured low PAHs EFs of 0.002 μg km$^{-1}$
during Artemis urban cold start and much higher values of PAHs up to 3.9 μg km$^{-1}$ during Artemis road, but surprisingly
none of the tested gasoline cars in the present work showed similar trends. Much higher values from 108 to 489 μg km$^{-1}$ during
WLTC cold start and from 27 to 102 μg km$^{-1}$ during WLTC hot start were recently reported for both gaseous and particle-
bound PAHs for three Euro 5 GDI cars (Muñoz et al., 2018). The PAHs EFs for the D4 car were 2.0 μg km$^{-1}$ for the cold urban
and 1.7 μg km$^{-1}$ for the motorway cycle. These findings are similar to the values of 2.3 μg km$^{-1}$ for a cold urban cycle and 0.6
μg km$^{-1}$ for a motorway cycle for a diesel Euro 5 measured by Alves et al. (2015). Vouitsis et al. (2009) reported PAHs EFs of
7.7 and 2.0 μg km$^{-1}$ for cold urban and motorway cycles, correspondingly, for a Euro 4 car equipped with a CDPF (which
actually corresponds to a Euro 5 technology). When Cheung et al. (2010) added a DPF to a Euro 4 diesel car (converting it
into a "Euro 5") the PAHs were below detection limit, underlining the high variability of PAHs emissions from different Euro
5 cars. As already mentioned in the previous sections both OA and PAHs EFs for D4 were surprisingly high for a diesel vehicle
equipped with a DPF suggesting a failure in the after-treatment device, as also supported by TEM images Figure S9 (b).

Sulfate, ammonium and nitrate EFs were generally low. The highest sulfate emissions were observed for the D3 during the
385 cold start WLTC cycle with 4.2 μg km$^{-1}$ and for the D1 during the Artemis motorway cycle with values of 1.3 μg km$^{-1}$. Both
D1 and D3 cars were equipped with a CDPF, while the D4 was equipped with an FBC-DPF and had lower sulfate EFs (0.18 -
0.22 μg km$^{-1}$), underlying the determining influence of the DPF technology on PM chemical composition. Nitrate was mostly





emitted by gasoline cars, the highest EFs were measured for the GDI3 car with 2.2 and 4.9 μg km$^{-1}$ for cold and hot start WLTC, respectively.

## 4 Conclusions

We characterized the chemical composition and we evaluated the emission factors of three diesel (two CDPF and one FBC-DPF) and four gasoline (three GDI and one PFI) Euro 5 light duty vehicles during transient cycles. Most of the particulate matter was emitted at the beginning of the cold start cycle due to the incomplete combustion and low catalyst efficiency. BC was always the dominant species accounting for 83-98% of the total particle mass concentration, while the corresponding OA fraction ranged between 1.8 and 14%. The OA emissions of the GDI gasoline cars were 5 to 12 times higher compared to the gasoline PFI vehicle OA emissions. Organosulfur containing ion fragments were detected for the first time at the exhaust of gasoline cars, probably from the release of lubricant oil, and accounted for 2-7% of the total organic mass concentration. In total 45 PAHs were identified and quantified. Similar to the OA, the PAHs emitted by the GDI car were considerably higher in comparison to those emitted by the PFI car. Approximately 52-66% of the PAHs were unsubstituted PAHs, followed by methylated PAHs (14-21% of the PAHs), oxygenated PAHs (5-19%), nitro-PAHs (1-11%) and amino PAHs (1-6%). Unexpectedly, the GDI car emitted the higher concentrations of nitro-PAHs, questioning the validity of using some NPAHs as marker of diesel emissions.

Oil droplets associated to metallic components as calcium, phosphorus, sulfur and zinc were also observed in PM from both gasoline and diesel vehicles. Analysis of particles emitted from the PFI vehicle revealed a disordered soot surface, which could affect the chemical reactivity and toxicity of the PM.

Nanoparticles in the size range of 15-20 nm, mainly composed of ammonium bisulfate, were measured during the motorway cycle, suggesting passive regeneration of the DPF for CDPF diesel vehicles. This behavior was not observed for the FBC-DPF vehicle, indicating that the different after-treatment strategy highly affects the PM size and composition.

Diesel cars equipped with well-functioning after treatment-devices generally emitted far less pollutants than the gasoline vehicles, but in the case of a DPF failure, very high levels of PM, similar to those reported for the GDI vehicles, were measured. This indicates that the DPF condition is important and special attention should be given to its maintenance during the lifetime of the vehicle. All particle characteristics investigated in this work should be taken into account in emission control strategies and in the assessment of the impact of light duty particle emissions on the environment and on human health.

*Data availability.* All data from this study are available from the authors upon request.

*Supplement.* The supplement related to this article is available on-line at: (link will be included by Copernicus).



*Author Contribution.* BD, MA and EK designed the research. EK performed and analyzed the measurements of the D4, PFI4
and GDI5 cars. AM-V performed and analyzed the measurements of the D3 and GDI3 cars. BR performed and analyzed the
measurements of the D1 and GDI1 cars. BM, BT-R, YL, CL, BV contributed to the experimental set up and the experimental
procedure. YL, CL and BV drove the cars. DF, CL, PP performed the TEM and XPS analysis. EK synthesized all the data and
wrote the paper with contributions from BD, DF and PP.

*Competing interests.* The authors declare that they have no conflict of interest.

*Acknowledgements.* The authors would thank for their helpful support of B. Boréave and L. Buriel from IRCELYON and P.
Tassel from IFSTTAR.

*Financial support.* This work was supported by the ADEME CORTEA program with the project CAPVEREA (contact
n.1466C0001) and the project MAESTRO (contract n. 1766C0001). The authors would like to thank the GDR SUIE for the
financial support to make complementary investigations on TEM and XPS analysis at the CINaM laboratory.

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





**Table 1: Technical characteristics of the tested vehicles, experimental conditions and particle phase instrumentation used.**

|  | Diesel Euro 5 | | | Gasoline Euro 5 | | | |
|---|---|---|---|---|---|---|---|
| **Vehicle name** | D1 | D3 | D4 | GDI1 | GDI3 | PFI4 | GDI5 |
| **Size class** | 1.5 DCI | 1.6 TDI | 2.0 HDI | 1.2 TSI | 0.9 TCE | 1.0 VVTI | 1.2 TCE 16 V |
| **Engine capacity (cm³)** | 1461 | 1598 | 1997 | 1197 | 900 | 998 | 1149 |
| **Weight (kg)** | 1090 | 1262 | 1515 | 1320 | 1055 | 1030 | 1100 |
| **Odometer mileage (km)** | 87073 | 78903 | 103000 | 25844 | 9500 | 27712 | 97089 |
| **Catalyst type** | DOC | DOC | DOC | TWC | TWC | TWC | TWC |
| **Particulate Filter type** | CDPF | CDPF | FBC-DPF | - | - | - | - |
| **Driving Cycles tested** | Artemis | WLTC | Artemis | Artemis | WLTC | Artemis | Artemis |
| **Dilution System** | CVS | FPS-4000 | Hot Injector | CVS | FPS-4000 | Hot Injector | Hot Injector |
| **Dilution Ratio** | 8-40 | 7-12 | 2.3-15 | 13-100 | 20-46 | 2.3 | 1.5 |
| **AMS type** | c-ToF | c-ToF | HR-ToF | - | c-ToF | HR-ToF | HR-ToF |
| **MAAP/AE-33** | MAAP | AE-33 | - | MAAP | AE-33 | - | AE-33 |
| **CPC** | YES | YES | - | YES | YES | YES | YES |
| **FMPS/SMPS/SMPS+E** | YES | YES | - | - | YES | - | - |
| **TEM/XPS** | YES/NO | YES/NO | YES/NO | YES/NO | YES/NO | YES/YES | - |








**Table 2: Correlations between mass spectra taken during the first two minutes of an Artemis motorway cycle and mass spectra from diesel and gasoline vehicles, lubricant oil and PMF factors related to fresh transportation (HOA, Hydrocarbon-like OA) from selected studies using both $R^2$ and the angle θ (Kostenidou et al., 2009) in parenthesis. The angle θ provides a better comparison for small difference in the mass spectra (where $R^2$ is less than 0.97).**

| | First 2 minutes of Artemis motorway ($R^2$ and angle θ in degrees) | | | | |
|---|---|---|---|---|---|
| | D1 | D4 | GDI3 | PFI4 | GDI5 |
| **Diesel bus exhaust[1]** | 0.80 (24.9) | 0.97 (9.1) | 0.88 (19.1) | 0.88 (19.8) | 0.92 (15.1) |
| **Diesel truck[2]** | 0.76 (27.4) | 0.95 (12.2) | 0.84 (21.8) | 0.83 (23.1) | 0.90 (16.8) |
| **Gasoline car[2]** | 0.74 (28.2) | 0.96 (11.2) | 0.83 (22.9) | 0.81 (25.7) | 0.91 (16.8) |
| **Diesel car Euro5[3]** | 0.82 (23.6) | 0.77 (26.9) | 0.83 (22.5) | 0.72 (29.2) | 0.73 (28.9) |
| **Lubricant oil (diesel car)[3]** | 0.75 (28.8) | 0.96 (12.3) | 0.86 (21.5) | 0.80 (27.9) | 0.89 (19.4) |
| **HOA Athens summer[4]** | 0.73 (29.3) | 0.95 (12.3) | 0.80 (24.8) | 0.83 (23.9) | 0.91 (16.8) |
| **HOA ICE-HT winter[5]** | 0.69 (31.9) | 0.94 (12.7) | 0.77 (26.8) | 0.80 (24.0) | 0.91 (16.8) |
| **HOA Barcelona[6]** | 0.77 (27.2) | 0.96 (10.8) | 0.85 (21.6) | 0.84 (24.3) | 0.92 (16.2) |

[1] Canagaratna et al. (2004), [2]Mohr et al. (2009), [3]R'Mili et al. (2018), [4]Kostenidou et al. (2015), [5]Kaltsonoudis et al. (2017), [6]Mohr et al. (2012)








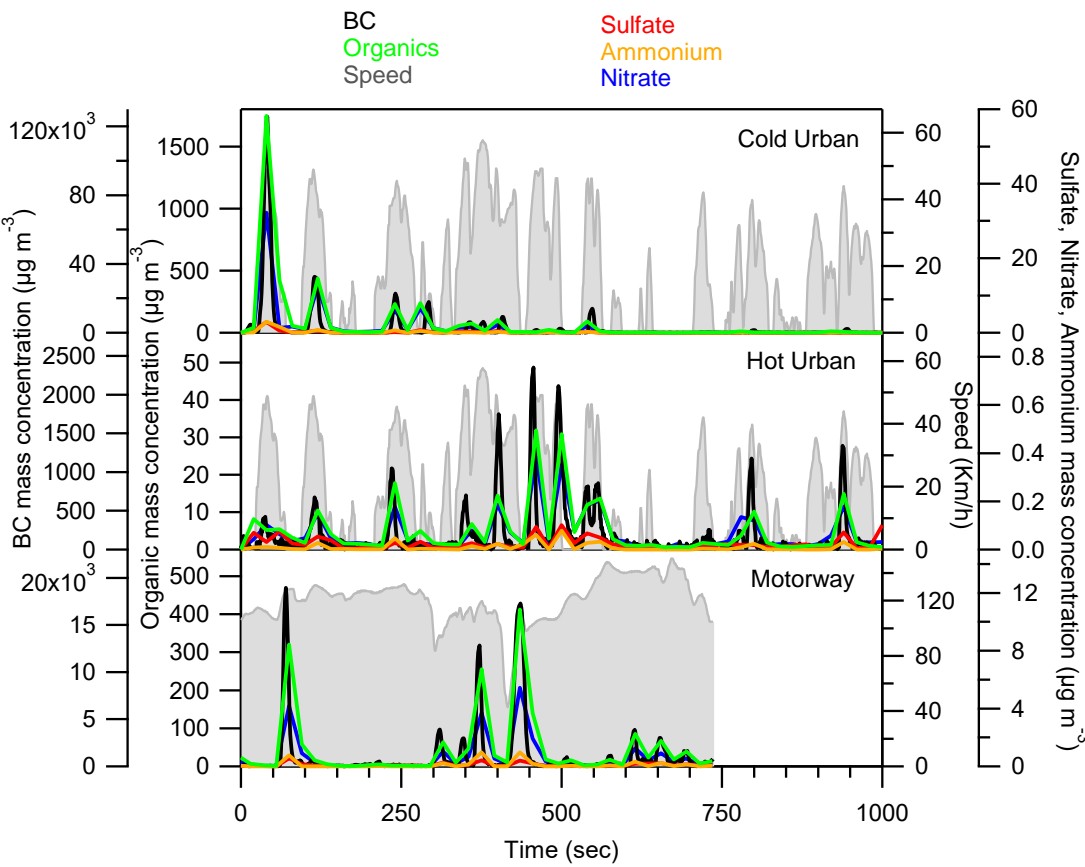

**Figure 1: OA, BC, sulfate, nitrate and ammonium time profiles during a cold urban, a hot urban and a motorway Artemis cycle for the GDI5 car.**

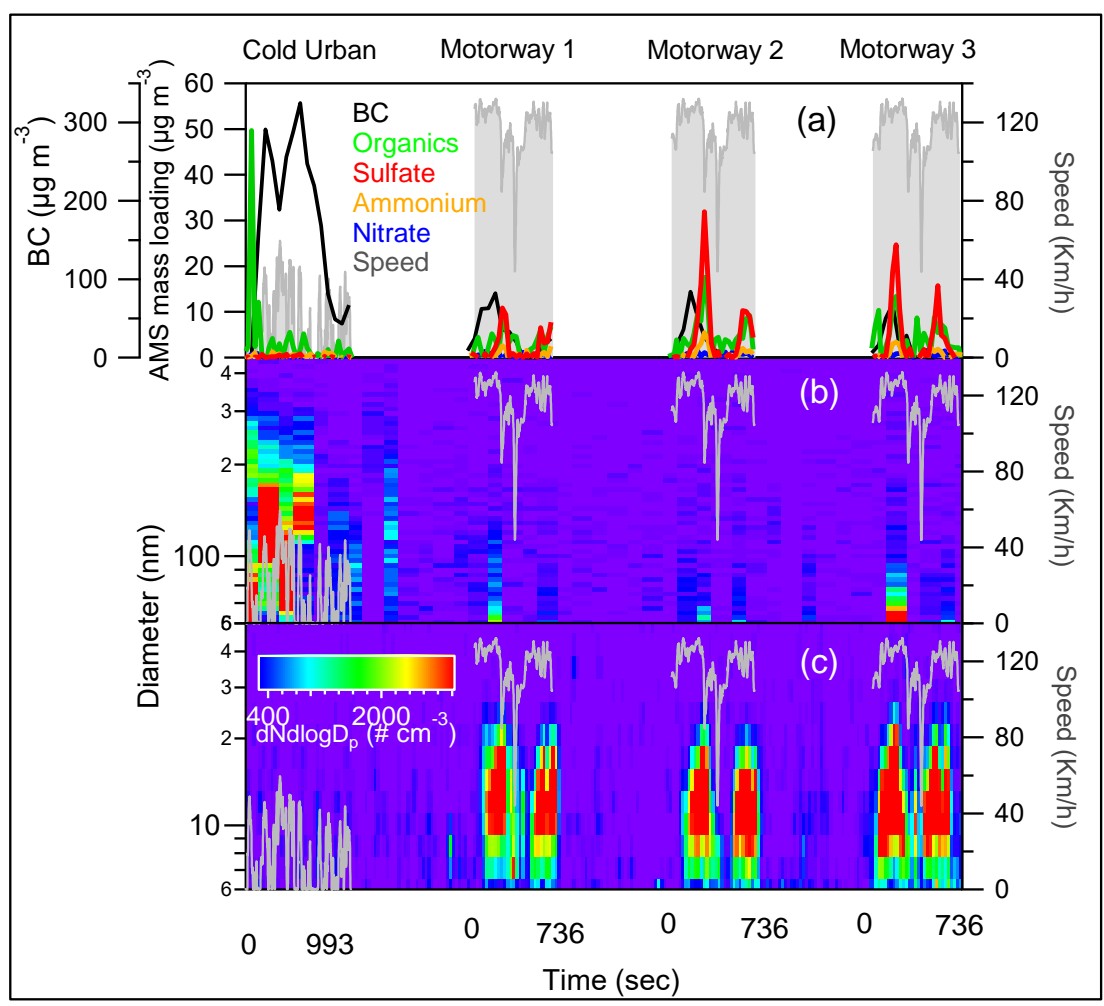

**Figure 2: PM emissions from the D1 vehicle, during a cold urban and 3 successive motorway cycles: (a) AMS major chemical species and BC (MAAP); (b) size distribution of fine particles (SMPS) and (c) size distribution of ultrafine particles (SMPS+E).**




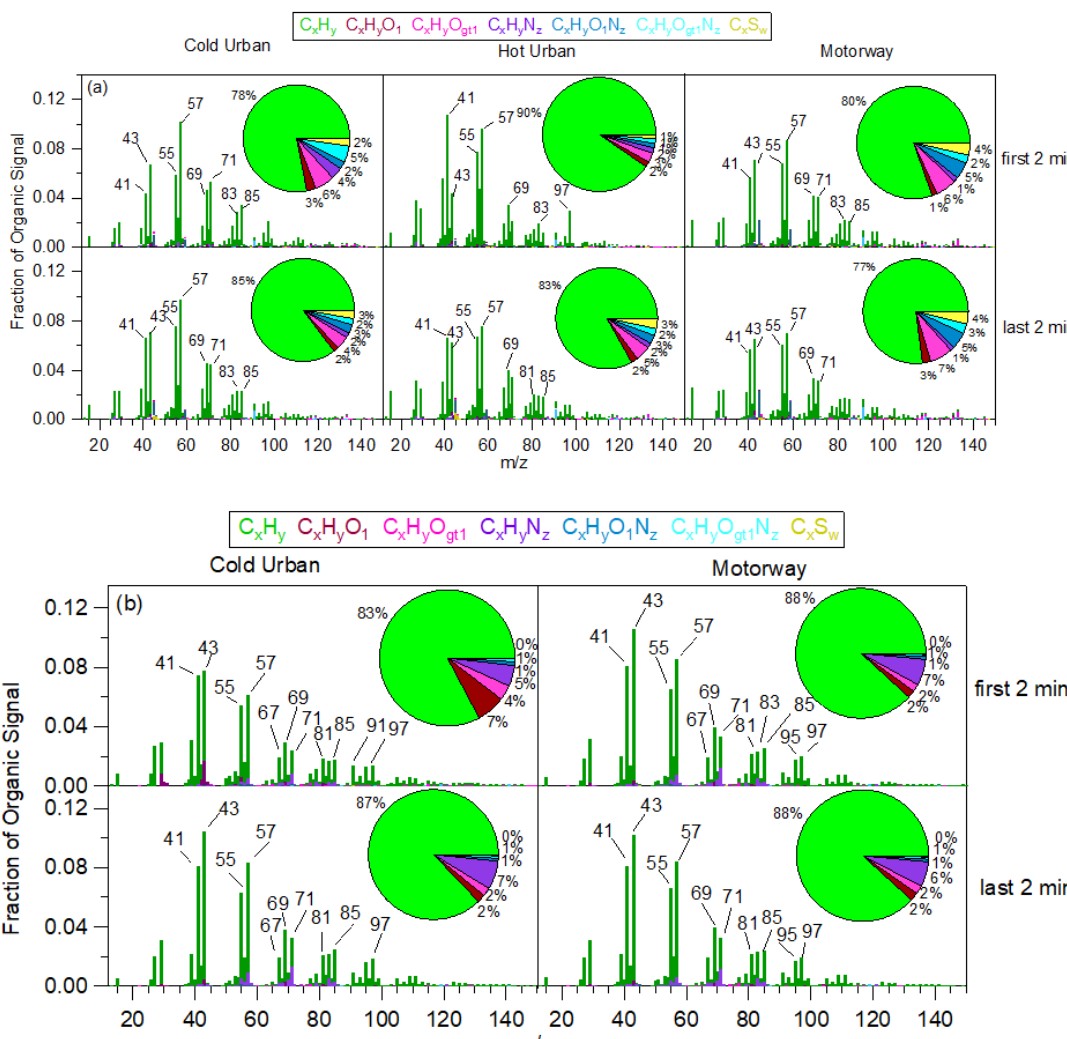

**Figure 3: HR mass spectra and chemical composition: (a) during the two first and the two last minutes of a cold urban (left), hot urban (middle) and a motorway (right) Artemis cycle for the gasoline GDI5 car and (b) during the two first and the two last minutes of a cold urban (left) and a motorway (right) Artemis cycle for the diesel D4 car.**







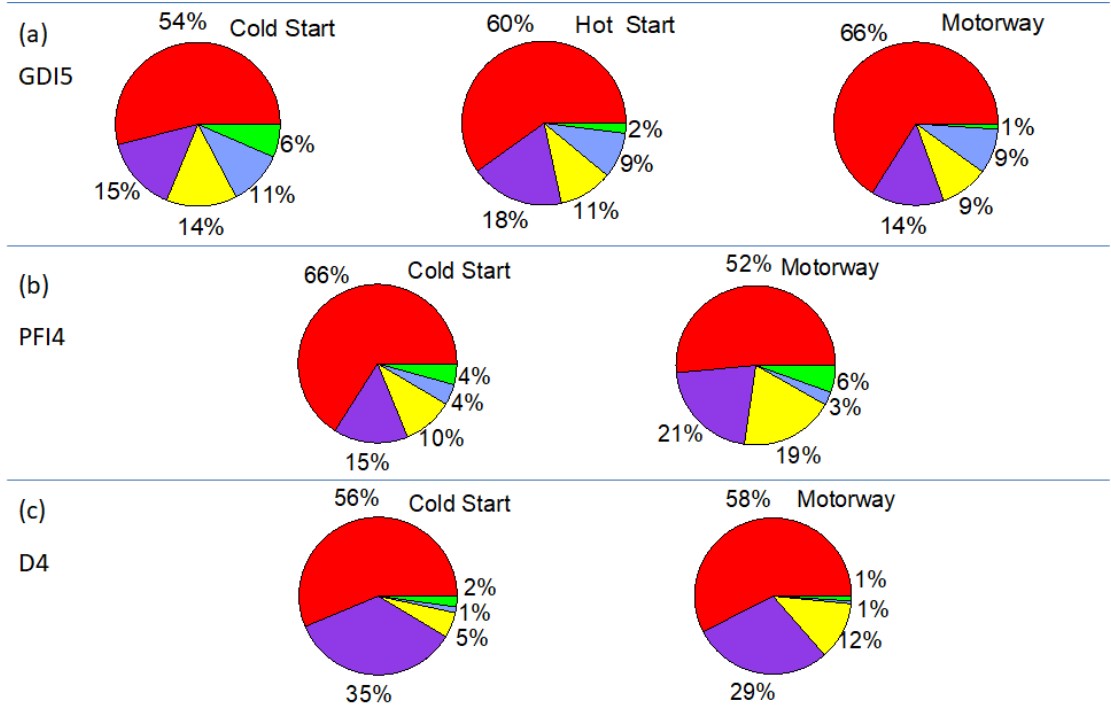


**Figure 4: Average contribution of the various PAHs families during Artemis cycles for GDI5 (a), PFI4 (b) and D4 (c) vehicles.**



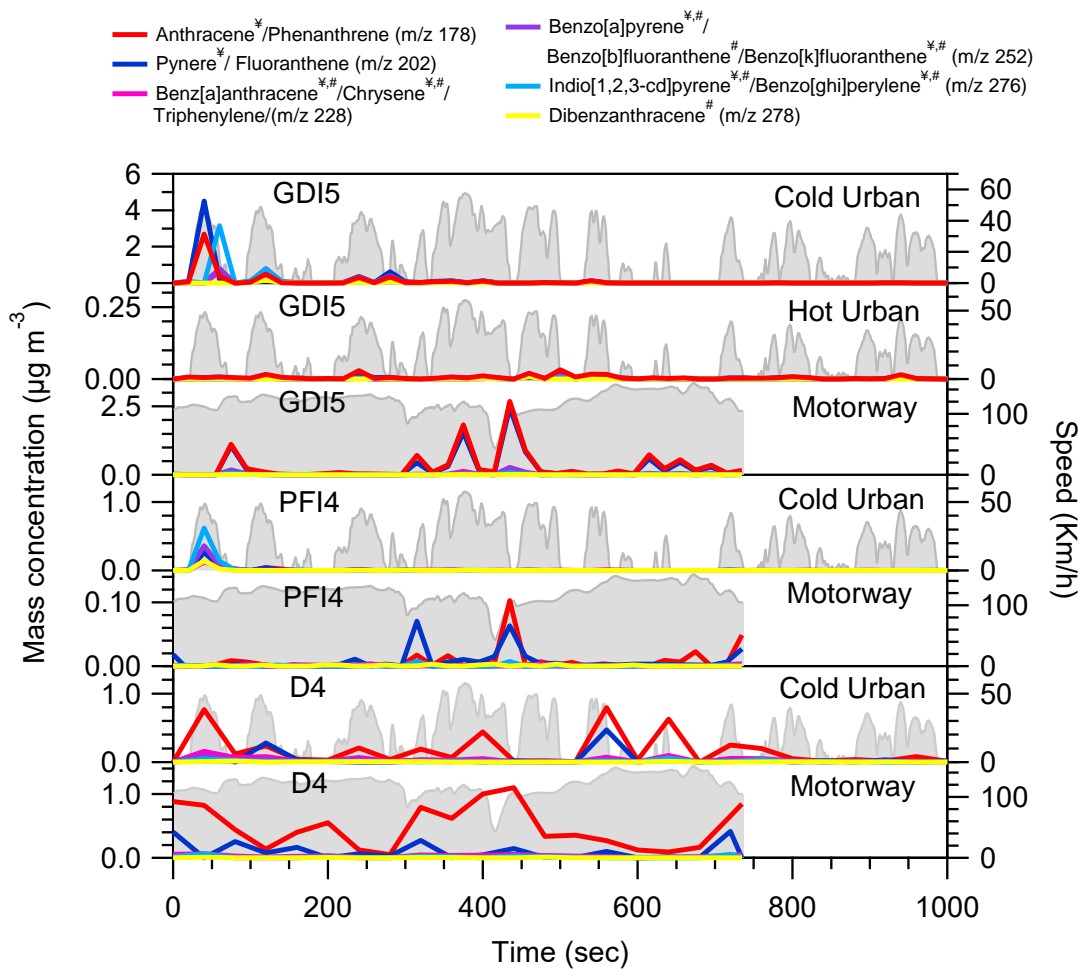

**Figure 5: Example of PAHs time series profiles for GDI5, PFI4 and D4 vehicles for Artemis tested cycles. The time series have been corrected for the injector dilution (1.5) and for a relative abundance $f_{A,i}$ =0.265. Compounds labelled with ¥ have been characterized as photomutagenic, while compounds labelled with # have been found to be cancerogenic according to WHO (2005).**





Figure 6: TEM images of samples collected during Artemis urban cold cycle: (a-c) GDI1 vehicle (dilution ratio 40), the sample was collected during the first 120 sec of the cycle; (d-f) GDI3 vehicle (dilution ratio 46), the sample was collected during the first 120 sec of the cycle; (g-i) D1 vehicle (dilution ratio 40) the sample was collected during the first 300 sec of the cycle.








**Figure 7: Emission factors for three diesel (D1, D3 and D4) and four gasoline (GDI1, GDI3, PFI4, GDI5) vehicles Euro5: BC EFs**
**are expressed in mg km⁻¹, while for organics, PAHS, sulfate, ammonium and nitrate the values are expressed in µg km⁻¹. Gasoline**
**cars are shown with solid bars and diesel cars with pattern bars. The error bars correspond to ±1σ standard deviation.**