# Peer review of "Technical Note: Emission factors, chemical composition and morphology of particles emitted from Euro 5 diesel and gasoline light duty vehicles during transient cycles"

_Atmospheric Chemistry and Physics, 2020_

## Referee Comment (RC1) · Anonymous Referee #1 · 11 Oct 2020

See Attached document.Review of Kostenidou et al. "Technical Note: Emission factors, chemical composition and morphology of particles emitted from Euro 5 diesel and gasoline light duty vehicles during transient cycles"

The manuscript details gas- and particle-phase emissions from 3 diesel vehicles and 4 gasoline light duty vehicles (1 PFI and 3 GDIs) certified to the Euro 5 emission standards. All vehicles were tested on a chassis dynamometer on 2 different driving cycles. Main conclusion from this study is that GDI engines emit more PM and gas-phase PAHs compared to a single PFI engine. The suite of instrument used measurements is com-

prehensive. The manuscript is well written and the measurements from this study are largely in-line with existing studies. However, given the small vehicle fleet size (and just a single PFI vehicle tested), I expect to see a more detailed discussion of the literature, especially for PFI gas and particle phase emissions. The authors often carefully report measurements of highly resolved instrumentations (e.g., PAHs, TEM XPS measurements) but fail to convey to the reader the main point (and the importance) of conducting these time consuming and expensive analyses. I recommend publication after the authors address these comments (and the below).

Minor comments: Line 24: Odd transition, both PAH and NPAH concentrations were higher in GDIs compared to the PFI engine.

Line 26: lower detection limit

Line 27: Emissions of Aitken-mode particles (particle with diameter < 30 nm)

Line 73. Citation needed when citing mentioning that NH3 might enhance NPF.

Lines 72-73: "Despite all the previous studies for the characterization of the vehicles' emissions, an integrated and comprehensive analysis for the understanding of these emissions is missing." Not sure what this means. Either be specific or remove.

Table1: Why is the numbering for PFI, GDI, and D vehicles not continuous. This is confusing, I suggest you adjust. There is a single PFI vehicle presented in the manuscript, yet it is numbered PFI4, leaving the reader the impression that the data from at least 4 PFIs were reported. Also, the use of the blue and red colors makes the table harder to read, I suggest keeping all text in black.

Line 94: add reference for the WLTC cycle. I also think that a brief description of each driving cycle is needed in section 2.1.

Line 98: "The line was heated at 80-120 C.." was this to reduce semi volatile losses?

Line 104: (Table 1)

Section 2.2.1: The authors do not discuss the CE of the AMS. Is it 1? 0.5? the choice of CE should be explicit in the manuscript. Also, have the authors compared the AMS non-refractory OM mass to filter OC mass? If OC filters were collected and analyzed the comparison between OC and OM should be provided in the MS or the SI.

Section 2.2.3: This section discusses TEM sampling and not "exhaust gas sampling". Please adjust.

Section 2.3 is named "instrumentation" so is section 2.2. Adjust

Line 140: "$CO_2$ (the signal at m/z 44)..."

Line 162: "the distance-based EF"

Line 164: are the concentrations $C(t)$ background corrected?

Line 165: What is DR? dilution ratio?

Line 170: BC is not measured by AMS. Adjust

Line 173: I was surprised to see that the BC fraction of PM emissions for GDI5 was so high for GDIs (>96%). The authors should provide how often this is seen in the literature.

Line 174: 120 $mg/m^3$ should be for BC and not organics.

Line 185: Refer to panels b and c in Fig 2 as Figure 2b and 2c. Right now, "(a) ... (b and c)" is confusing. Adjust also in other instances in the text.

Lines 190 and 196: The authors claim that the sulfate measured is in ammonium bisulfate form yet offer no justification to why that is. Either justify your assumption or remove.

Line 207: "Figure 3 shows the HR-AMS mass spectra for the GDI5 and the D4 vehicle during the first (cold start?) and last 2 minutes (hot running exhaust?) of each cycle."

Line 228: "Sulfur containing ion fragments were mostly emitted from hot engines (end

of urban cycle and motorway cycle) and are tentatively explained by the release of some lubricant oil." Why would this be the case for the GDI5 vehicle and not for the D4 vehicle as well? It is clear from Figure 3 that only the GDI5 showed trace amounts of sulfur containing organic fragments

Figure 3: It is worth mentioning in the text that there are no significant differences in the non-refractory organic composition of exhaust particle emissions for cold start vs hot start.

Line 261: The author mention that MPAH are carcinogenic compounds. Looking at Table S3 in the SI, it seems that MPAH emissions during hot start (the large fraction of a trip) are usually larger than during cold starts which can have important implication on public health near emission sources.

Line 276: replace "the car" with "vehicle tailpipe"

Line 293: "(below 20 nm)". Give exact statistics (mean, std). This data is helpful for studies looking at the optical properties of fresh combustion soot particles.

Line 306: The authors assume that inclusions in soot particles are metals. The authors should provide a reference for this claim.

Lines 331-346: it is not clear to me what are the main points the authors are trying to make from that large paragraph.

Line 412: replace "should be taken into account"

Figure 1: Change OA in caption to organics

Figure 3: make font in panel a same as panel b.

[Figure]

---

## Referee Comment (RC2) · Anonymous Referee #2 · 2 Nov 2020

The manuscript acp-2020-842 by Kostenidou et al. presents a relatively comprehensive analysis of emissions from Euro 5 diesel and GDI vehicles. The work is well done, thoroughly discussed, and presents useful data. However, the manuscript is possibly more suited to a journal focussed on emissions and air quality, as it represents incremental progress. However, ACP has published similar work before. This decision is ultimately the Editor's. However, considering the thorough analysis and useful data, I would recommend publication in ACP after the following revisions.

[Figure]

**Major comments**

1.

The manuscript is not a technical note in my opinion. The analysis and literature discussion are thorough. There are no new technical advances. The label does not seem appropriate.

2.

The abstract, manuscript, and figures do not emphasize enough that the diesel engines used DPFs while the gasoline engines did not. Of course particulate emissions were lower after the DPFs.

Line 14 should be changed to "BC ... emission factors varying from 0.2 to 7.1 mg/km for gasoline cars and 0.003 to 0.08 mg/km for diesel vehicles with DPF". Every other statement comparing the two should include "with DPF". Scientifically, the manuscript is effectively reporting the efficacy of the DPF by comparison with the gasoline case. However, this does make sense since the vehicles were all commercially available.

3.

This manuscript relies heavily on the methodology of Herring et al. (2015) to estimate speciated PAH concentrations. However, the Herring et al. 2015 metholodogy is not analytically reliable. The authors did not use any laboratory standards, and had no way of knowing whether PAHs had fragmented prior to their identification. Therefore, major potential issues, such as the relative sensitivity of the method to PAHs with different functional groups (which are extremely likely to fragment during electron ionization) were not explored.

Herring et al. (2015) only presented validation data from a photoelectric aerosol sensor, which is another indirect method of PAH quantification. If assumed accurate, the validation data from the photoelectric sensor imply that the accuracy of the AMS technique is at best 210% (calculated from their correction factors of 67 and 208).

Therefore, the published mass spectral interpretation of Herring et al. (2015) should not be taken as an analytical standard and should not be used in routine engine emissions analysis until further validation work is completed. The current manuscript presents misleading results, as most readers will assume that the authors used reliable methods such as GCMS when seeing the present manuscript.

The authors should delete Figure 4 and simplify Figure 5 to represent only "PAH" and "functionalized PAH" from the AMS data sets. If the authors wish to keep Figure 4 in the SI, they should add a comment to the caption to emphasize that the ersults are only estimated.

**Minor comments**

Line 42: Dallmann et al. 2014 also belongs in this list.

Line 90: Please briefly mention why 3 different dilution systems were used, and what differences could be expected as a result.

Line 142: Is conductive silicone really present in the tailpipe, or in the sampling system? The Timko citations did not discuss tailpipe tubing.

Line 215: "we conclude that both GDI5 and D4 emitted randomly oil droplets". Surely the emissions are not random, and surely the authors understand a little more than they are saying here. Do the authors mean that the lubrication oil emissions were not correlated with engine load, cycle period, etc.? Please improve the statement to represent a better scientific discussion.

Table 2: Define theta. R = cos(theta in degrees). The text suggests that theta is a different parameter to R, when it is not.

Figure 4: OPAH is not readable in light yellow text.

Figure 6: Add interpretation to the caption. I also suggest adding one of the micrographs from Figure S9 which show the tiny metallic particles, since this is a rare observation.

Data availability: I recommend that the authors upload a table with the data of Figure 7 instead of stating that the data are available on request.

---

## Author Comment (AC1) · 31 Jan 2021

Referee #1:

The manuscript details gas- and particle-phase emissions from 3 diesel vehicles and 4 gasoline light duty vehicles (1 PFI and 3 GDIs) certified to the Euro 5 emission standards. All vehicles were tested on a chassis dynamometer on 2 different driving cycles. Main conclusion from this study is that GDI engines emit more PM and gas-phase PAHs compared to a single PFI engine. The suite of instrument used measurements is com-

prehensive. The manuscript is well written and the measurements from this study are largely in-line with existing studies. However, given the small vehicle fleet size (and just a single PFI vehicle tested), I expect to see a more detailed discussion of the literature, especially for PFI gas and particle phase emissions. The authors often carefully report measurements of highly resolved instrumentations (e.g., PAHs, TEM XPS measurements) but fail to convey to the reader the main point (and the importance) of conducting these time consuming and expensive analyses. I recommend publication after the authors address these comments (and the below).

We added to the 4th paragraph of the introduction more literature work on cars technology (Wang et al., 2016; di Rattalma and Perotti, 2017; Chan et al., 2014; Short et al., 2015). We also expanded the 4th paragraph of the introduction as: "Despite many studies investigated vehicles' emissions in the past, non-regulated pollutants are still not well identified and quantified for recent car's technologies. European databases (as COPERT 5) are still missing information on PM chemical composition preventing a full assessment of car emissions on urban air quality and health. This work provides a detailed description and comparison of PM emissions in terms of chemical composition and emission factors from three diesel and four gasoline (both GDI and PFI) commercial Euro 5 light-duty vehicles tested on a roll bench chassis dynamometer facility. The chemical speciation of particulate matter included BC, organics, sulfate, ammonium, nitrate and PAHs and was accomplished using real time and high-resolution instrumentation, providing real-time emission profiles and information on the impact of after-treatment devices along the cycle. Particles morphology, surface structure and elemental composition were furthermore investigated using offline analysis." These time consuming and expensive analyses reveal new features of cars pollutants: as the functionalized PAHs and metallic nanoparticles and the potential health related impacts.

Minor comments: Line 24: Odd transition, both PAH and NPAH concentrations were higher in GDIs compared to the PFI engine.

Yes, it is true. We synthetized the sentence as follows: "Both PAHs and nitro-PAHs

emissions from the GDI technology were more than a factor of 4 higher compared to the vehicles equipped with a PFI system during the cold start cycle"

Line 26: lower detection limit.

We replaced "close to" with "below".

Line 27: Emissions of Aitken-mode particles (particle with diameter < 30 nm).

Aitken-mode particles are particles with diameters between 20nm and 100nm, so the particles below 30 nm cannot be considered as Aitken-mode particles. In the reality those particles were 15 nm and so we replaced "Emissions of nanoparticles (below 30 nm)" with "Emissions of particles around 15 nm".

Line 73. Citation needed when citing mentioning that NH3 might enhance NPF.

We added some citations from the following articles: P. Korhonen et al., 1999, Ortega et al., 2008; Pikridas et al., 2012; Kürten, 2019.

Lines 72-73: "Despite all the previous studies for the characterization of the vehicles' emissions, an integrated and comprehensive analysis for the understanding of these emissions is missing." Not sure what this means. Either be specific or remove.

As mentioned before, we changed this part as into: "Despite many studies investigated vehicles' emissions in the past, non-regulated pollutants are still not well identified and quantified for recent car's technologies. European databases (as COPERT 5) are missing information on PM chemical composition preventing a full assessment of car emissions on urban air quality and health. This work provides a detailed description and comparison of PM emissions in terms of chemical composition and emission factors from three diesel and four gasoline (both GDI and PFI) commercial Euro 5 light-duty vehicles tested on a roll bench chassis dynamometer facility. The chemical speciation of particulate matter included BC, organics, sulfate, ammonium, nitrate and PAHs and was accomplished using real time and high-resolution instrumentation, providing real-time emission profiles and information on the impact of after-treatment devices along

the cycle. Particles morphology, surface structure and elemental composition were furthermore investigated using offline analysis."

Table1: Why is the numbering for PFI, GDI, and D vehicles not continuous? This is confusing, I suggest you adjust. There is a single PFI vehicle presented in the manuscript, yet it is numbered PFI4, leaving the reader the impression that the data from at least 4 PFIs were reported. Also, the use of the blue and red colors makes the table harder to read, I suggest keeping all text in black.

The reviewer is right there was a mistake in the numbering, which was based on the whole set of cars measured, but for this paper we selected EURO5 cars only. We changed the numbering as follow:

D1–> D1

D3–> D2

D4–> D3

GDI1–> GDI1

GDI3–> GDI2

PFI4–> PFI

GDI5–> GDI3

We changed the color in Table 1, turning everything in black.

Line 94: add reference for the WLTC cycle. I also think that a brief description of each driving cycle is needed in section 2.1.

The cycles are well known and therefore we refer to literature for their description. We reformulated the sentence as follows: "The emissions of vehicles D3 and GDI3 (D2 and GDI2 with the new numeration) were tested using the Worldwide Harmonized Light Vehicles Cycle (WLTC), (Tutuianu et al., 2015), which is the official cycle for emissions

legislation of Euro 6 cars, while the vehicles. . ..."

Line 98: "The line was heated at 80-120 C.." was this to reduce semi volatile losses?

Yes, the reviewer is right, we had simultaneous measurements of VOCs, IVOCs and SVOCs. We added the sentence: "The line was heated at 80-120o C in order to reduce losses of the semi-volatile compounds (SVOCs) which were analyzed by PTR-MS and GC-MS techniques (Marques et al., in preparation)."

Line 104: (Table 1).

We put the parenthesis before the word Table.

Section 2.2.1: The authors do not discuss the CE of the AMS. Is it 1? 0.5? the choice of CE should be explicit in the manuscript. Also, have the authors compared the AMS non-refractory OM mass to filter OC mass? If OC filters were collected and analyzed the comparison between OC and OM should be provided in the MS or the SI.

The CE selection was discussed in the 2.3.1 section (Organics and PAHs). We used a CE of 1 following previous studies for engine exhaust emissions which used a CE of 1 (Canagaratna et al., 2004; Slowik et al., 2004; Dallmann et al., 2014; Eriksson et al., 2014; Bruns et al., 2015; Herring et al., 2015). Unfortunately, we did not collected filters for total OC measurements so a direct comparison between filters and AMS cannot be done.

Section 2.2.3: This section discusses TEM sampling and not "exhaust gas sampling". Please adjust.

We changed the title into "Off-line analysis: TEM-EDX and XPS techniques".

Section 2.3 is named "instrumentation" so is section 2.2. Adjust.

We changed the title into "Methods".

Line 140: "CO2 (the signal at m/z 44): : :"

We put a comma after the word CO2, so that the sentence becomes clear. The signal at m/z 44 in AMS could have either gaseous or particulate origin. In our case (fresh vehicle emissions) the vast majority of this signal is deriving from gaseous $CO_2$ (rather than from particulate oxygenated compounds), thus it was totally removed from the fragmentation table in order to avoid artifacts in the organic mass concentration.

Line 162: "the distance-based EF".

We added the characterization distance-based before EF.

Line 164: are the concentrations C(t) background corrected?

The concentrations C(t) were indeed corrected subtracting the background concentrations when there was only clean air passed. These concentrations were practically zero. We added in the text that the C(t) concentrations were background corrected.

Line 165: What is DR? dilution ratio?

DR is the external dilution ratio. We updated the manuscript.

Line 170: BC is not measured by AMS. Adjust.

We reformed the sentence into: "Figure 1 shows the particle mass concentration transient profile of the BC recorded by the Aethalometer and those of, organics, sulfate, nitrate and ammonium measured by the HR-ToF-AMS..."

Line 173: I was surprised to see that the BC fraction of PM emissions for GDI5 was so high for GDIs (>96%). The authors should provide how often this is seen in the literature.

Indeed, this high BC fraction has been observed in previous studies for GDI Euro 5 vehicles (e.g., Platt et al., 2017; Pieber et al., 2018), who found that the BC emission factors during cold WLTC starts were one order of magnitude higher compared to the organic emissions factors. In these studies, BC was around 80-90% of the BC+OA sum. Later in the manuscript (section 3.3 on emissions factors) we already provide

a direct comparison between our BC and OA EFs to the corresponding EFs of those papers. We decided to compare directly the EFs rather than the percentages. To our knowledge we did not find many other papers providing both BC and organics EFs for GDI Euro 5 vehicles.

Line 174: 120 mg/m3 should be for BC and not organics.

The reviewer is right: 120 mg m-3 refers to BC while and 1600 $\mu$g m-3 to organics. We correct the above sentence.

Line 185: Refer to panels b and c in Fig 2 as Figure 2b and 2c. Right now, "(a) : : : (b and c)" is confusing. Adjust also in other instances in the text.

We reformulated the sentence into: "Figure 2 shows the PM emissions of the D1 car (equipped with a CDPF) in terms of chemical composition (Figure 2a) and particle size distribution (Figures 2b and 2c) during a cold urban and three consecutive motorway cycles."

Lines 190 and 196: The authors claim that the sulfate measured is in ammonium bisulfate form yet offer no justification to why that is. Either justify your assumption or remove.

We claim that the sulfate measured is in the form of ammonium bisulfate, because of the mass ratio of the sulfate/ammonium: the measured mass concentration ratio of the sulfate/ammonium was around 5, which means it is rather ammonium bisulfate, which has a sulfate/ammonium mass ratio equals to 5.33. In case of ammonium sulfate the corresponding ratio would be close to 2.67. We added a clarification that the assumption of ammonium bisulfate is based on the sulfate/ammonium mass concentration ratio: "The identification of ammonium bisulfate was based on the sulfate/ammonium mass concentration ratio."

Line 207: "Figure 3 shows the HR-AMS mass spectra for the GDI5 and the D3 vehicle during the first (cold start?) and last 2 minutes (hot running exhaust?) of each cycle."

We assume that the reviewer means D4 instead of D3 as Figure 3 contains the MS of the GDI5 and D4; with the new numeration these cars correspond to GDI3 and D3. These mass spectra of Figure 3a refer to the two first and two last minutes of each cycle (cold urban, hot urban and motorway) of the GDI car. This means only the 2 first minutes of the cold cycle correspond to a cold start, while all the remaining spectra correspond to hot engine conditions. The engine is "really cold" only during the first cold start of the day, while after that first cycle, the hot urban or the motorway will start with a warm/hot engine (pre-heated). Thus, the beginning of the hot and motorway cycles is characterized by hot engine conditions (this is a very well-known procedure in engine tests). In order to avoid any confusion, we reformulated the sentence as follows: "Figure 3 shows the HR-AMS mass spectra for the GDI3 and the D3 vehicle during the first two and last two minutes of each type of cycle (cold urban, hot urban and motorway)."

Line 228: "Sulfur containing ion fragments were mostly emitted from hot engines (end of urban cycle and motorway cycle) and are tentatively explained by the release of some lubricant oil." Why would this be the case for the GDI5 vehicle and not for the D3 vehicle as well? It is clear from Figure 3 that only the GDI5 showed trace amounts of sulfur containing organic fragments.

Again, we assume that the reviewer means D4 instead of D3 as Figure 3 contains the MS of the GDI5 and D4; with the new numeration these cars correspond to GDI3 and D3. Figure 3a shows that the higher contribution of sulfur containing fragments are observed at the end of the cold urban, at the end of the hot urban and at both the beginning and the end of the motorway cycle of the GDI5 (GDI3 with the new numeration). This shows that these compounds are emitted when the engine is hot. On the contrary Figure 3b, which refers to the D3 car, shows that the contribution of the sulfur containing fragments was 0% either the engine was cold or hot. Thus, based on the HR mass spectra speciation, D3 car does not emit sulfur containing fragments.

Figure 3: It is worth mentioning in the text that there are no significant differences in

the non-refractory organic composition of exhaust particle emissions for cold start vs hot start.

At the line 230: we added the following sentence: "Except for the above differences (presence of oil droplets and sulfur containing fragments) no other significant variability was observed between the cold and the hot start HR mass spectra".

Line 261: The author mention that MPAH are carcinogenic compounds. Looking at Table S3 in the SI, it seems that MPAH emissions during hot start (the large fraction of a trip) are usually larger than during cold starts which can have important implication on public health near emission sources.

The authors suggest to the reviewer to look at Figure 4 instead of Table S4 which presents some of the major PAHs (not all the PAHs). On the other hand, Figure 4 shows all the PAHs families in % and from this figure it is not that clear the increase of MPAHs during hot cycles. This is the case for the PFI car, but it is not the case for the diesel. For the GDI vehicle the corresponding fraction is very alike between the 3 cycles. We do prefer at that stage to avoid a general conclusion. In our future publication on Euro 6 cars, we will discuss more in details PAHs speciation since we could collect many more filters in parallel to AMS analysis and it would be easier to have a more conclusive statement on the trend of MPAHs and NO2PAHs vs UnsubPAHs. We will also have the opportunity to analyze NO2PAHs collected on filters.

Line 276: replace "the car" with "vehicle tailpipe".

We make the above replacement.

Line 293: "(below 20 nm)". Give exact statistics (mean, std). This data is helpful for studies looking at the optical properties of fresh combustion soot particles.

For the cold cycle the gasoline primary particles had a mean diameter of 24.2±4.1 nm and during the hot cycles on mean diameter of 14.5±3.4nm (these values have been added to the text).

Line 306: The authors assume that inclusions in soot particles are metals. The authors should provide a reference for this claim.

The paragraph has been revised and literature work added. "Figure 6 (j-l) depicts soot particles from the D3 car tiny sparse dark spots were ubiquitous within the soot particles and were interpreted as metal inclusions. Unfortunately, EDX could not reveal their chemical nature due to the very small amount of material in these inclusions as they were very small (typically less than 0.5 nm) and their spatial density was low. Nevertheless, we suppose that these inclusions were metallic and resulted from the after-treatment device of the FBC-DPF vehicle (D3), which use metallic salts or organometallic additives into the engine combustion chamber. Upon combustion, the additive produces nanoparticles of metal oxides that are mixed with soot particles and get deposited on the DPF walls. The role of these metals is to reduce the DPF regeneration temperature (Ntziachristos et al., 2005; Majewski and Khair, 2006; Song et al., 2006). During DPF regeneration soot oxidation takes place, the DPF filtration efficiency is consequently reduced, (R'mili et al. 2018) allowing for few minutes the emission of the ash particles attached on/or enclosed in soot may occur (Liati et al., 2018)".

Lines 331-346: it is not clear to me what are the main points the authors are trying to make from that large paragraph.

The point was to describe the nature of the soot surface. Indeed surface defects and bound oxygen may imply higher reactivity and possible health implication of soot particles.

Line 412: replace "should be taken into account".

We replaced it with "should be considered".

Figure 1: Change OA in caption to organics.

We replaced "OA" with "Organics".

Figure 3: make font in panel a same as panel b.

We changed the font in panel a.

References:

Bruns, E. A., Krapf, M., Orasche, J., Huang, Y., Zimmermann, R., Drinovec, L., Močnik, G., El-Haddad, I., Slowik, J. G., Dommen, J., Baltensperger, U., and Prévôt, A. S. H.: Characterization of primary and secondary wood combustion products generated under different burner loads, Atmos. Chem. Phys., 15, 2825–2841, 2015.

[revised manuscript text omitted]

---

## Author Comment (AC2) · 31 Jan 2021

Referee 2:

The manuscript acp-2020-842 by Kostenidou et al. presents a relatively comprehensive analysis of emissions from Euro 5 diesel and GDI vehicles. The work is well done, thoroughly discussed, and presents useful data. However, the manuscript is possibly more suited to a journal focused on emissions and air quality, as it represents incremental progress. However, ACP has published similar work before. This decision is

ultimately the Editor's. However, considering the thorough analysis and useful data, I would recommend publication in ACP after the following revisions.

Major comments

1.The manuscript is not a technical note in my opinion. The analysis and literature discussion are thorough. There are no new technical advances. The label does not seem appropriate.

The authors fully agree with the reviewer, this is a research article, and indeed we sent it to ACP as a research paper. The categorization as "technical note" has been suggested by an Executive Editor of ACP, we could not do much about it.

2. The abstract, manuscript, and figures do not emphasize enough that the diesel engines used DPFs while the gasoline engines did not. Of course, particulate emissions were lower after the DPFs.

Line 14 should be changed to "BC ... emission factors varying from 0.2 to 7.1 mg/km for gasoline cars and 0.003 to 0.08 mg/km for diesel vehicles with DPF". Every other statement comparing the two should include "with DPF". Scientifically, the manuscript is effectively reporting the efficacy of the DPF by comparison with the gasoline case. However, this does make sense since the vehicles were all commercially available.

We did not emphasize the presence of the DPF on the diesel cars and the absence GPF in the gasoline cars, because in our study we examined EURO 5 vehicles only. By default, diesel EURO 5 cars are equipped with a DOC+DPF while gasoline EURO 5 vehicles are not equipped by any particle filtration system. These information are provided in Table 1.

We modified section 2.1. about vehicles description as follows: "We tested three GDI cars and one PFI vehicle all equipped with three-way-catalysts (TWC), while all diesel cars were equipped with a Diesel Oxidation Catalyst (DOC) and a Diesel Particulate Filter (DPF); two were catalyzed (CDPF) and one was fuel borne catalyst (FBC-DPF)".

We think that this would help the reader to remind that gasoline cars are equipped with TWC only and not DPF.

In the introduction we have this additional sentence talking about recent reduction of diesel sales: "This trend continues despite the very efficient PM (and BC) reduction achieved by diesel cars, due to the combination of DOC and DPF (Gordon et al., 2013; Platt et al., 2017)".

The authors do not think that they should repeat that diesel had DPF along the entire article, but it is true that we can stress it in the discussion and conclusion.

Furthermore, it is worth to keep in mind that if the DPF has some issues, as this work highlights with the D3 diesel car, then its emissions can be as high as those observed for GDI cars (Figure 7). Thus, the presence of the filter alone cannot characterize the level of the emissions, DPF (or GPF) has to work properly.

3.This manuscript relies heavily on the methodology of Herring et al. (2015) to estimate speciated PAH concentrations. However, the Herring et al. (2015) methodology is not analytically reliable. The authors did not use any laboratory standards, and had no way of knowing whether PAHs had fragmented prior to their identification. Therefore, major potential issues, such as the relative sensitivity of the method to PAHs with different functional groups (which are extremely likely to fragment during electron ionization) were not explored. Herring et al. (2015) only presented validation data from a photoelectric aerosol sensor, which is another indirect method of PAH quantification. If assumed accurate, the validation data from the photoelectric sensor imply that the accuracy of the AMS technique is at best 210% (calculated from their correction factors of 67 and 208). Therefore, the published mass spectral interpretation of Herring et al. (2015) should not be taken as an analytical standard and should not be used in routine engine emissions analysis until further validation work is completed. The current manuscript presents misleading results, as most readers will assume that the authors used reliable methods such as GCMS when seeing the present manuscript.

[Figure]

The reviewer is right about the need of an intercomparison with GC-MS analysis or other reference methods. Unfortunately, due to an overlapping of issues (COVID lockdown, shutdown of instrumentation in the laboratory) we were not able to finalize the work before the deadline and we did not have many filters available taken at the same time of the on-line analysis (AMS). Once we got access to the laboratory again, we used few quartz filters sampled during one campaign and we compared GC-MS analysis for GDI3 car with AMS results.

Results about this intercomparison are summarized in table S3 in the supplementary material. GC-MS calibration was carried out using the following standards (phenanthrene, fluorine, fluoranthene, benzo[k]fluoranthene, benzo[a]pyrene and benzo[a]pyrene, methyl-naphthalene and anthraquinone).

Higher concentrations are detected by AMS (following Dzepina et al. (2007) and Herring et al. (2015) methods) with respect to GC-MS analysis for light unsubstituted PAHs and alkyl-PAHs good agreement for Oxy-PAHs and underestimation is observed for heavy unsubstituted PAHs (as benzopyrene and isomers). NitroPAHs were not analyzed by GC-MS.

We remind that the sampling line was heated at 80-120°C to avoid losses of semi-volatile VOCs (paper in preparation). For particle phase we did used a split of the main line in the last section (3 meters) which was at room temperature. The AMS has a sampling flow below 100ml/min and this distance was enough to get a cold sample. The particles collected on the filters were sampled at 20 l/min, and we are not sure that room temperature was achieved when the flow impacted the filters. The authors think that some volatilization issue for these samples may have occurred.

Some explanation about the methods. Herring et al. (2015) method is based on the paper of Dzepina et al. (2007), which showed that the AMS has the capability to quantify particle bound PAHs in real time in the ambient atmosphere. Dzepina et al. (2007), showed that unsubstituted and methyl-PAHs undergo little fragmentation during

ionization in the AMS, and that the AMS results agreed well both with photoelectric aerosol sensor (PAS) measurements and also GC-MS offline filter samples for several PAHs and few methyl-PAHs. The results of the intercomparison gave an uncertainty of +35% and −38%. More recently Hartikainen et al. (2020) applied Herring et al. (2015) method to PAHs in fresh and aged residential wood combustion emissions. It is true that for alkyl- and nitro-PAHs more fragmentation is expected.

The authors consider that Herring et al. (2015) is a quite reliable method for PAHs identification and estimation with the advantage of being an online technique, issue remains for nitro-PAHs and some alkyl-PAHs. The authors will be careful and will rather use the term estimate and identification instead of quantification.

A recent work from Yang et al. (2018) measured PAHs and functionalized PAHs both in the gas phase and particle phase emitted from GDI cars. The authors used Teflon filters for particle bound PAHs and used GC/MS. For NO2-PAHs they used CI negative mode GC-MS. Their results are discussed in the EFs PAHs and are, in general, in good agreement to what found in our work. Surprisingly low NO2-PAHs were found by Yang et al. (2018) for GDI vehicles (1% of the total PAHs).

The authors should delete Figure 4 and simplify Figure 5 to represent only "PAH" and" functionalized PAH" from the AMS data sets. If the authors wish to keep Figure 4 in the SI, they should add a comment to the caption to emphasize that the results are only estimated.

The authors prefer to keep both Figure 4 and Figure 5 in the main paper as they show the fraction and the profile during the cycles (two different messages). We explained in each figure caption that these results are based on the method of Herring et al. (2015).

Minor comments

Line 42: Dallmann et al. 2014 also belongs in this list.

We added Dallmann et al. (2014) in this list.

[Figure]

Line 90: Please briefly mention why 3 different dilution systems were used, and what differences could be expected as a result.

There were three different dilution systems used because the data presented in this paper were taken from 4 different campaigns conducted in different years and thus each set up was different. We have already mentioned this in section 2.2 (Instrumentation). However, we added a sentence in this place also in order to emphasize it. We do not expect to see any differences in our results, using those 3 different systems regarding the pollutants we measured (mass concentration of organics, sulfate, ammonium, nitrate, PAHs and BC).

Line 142: Is conductive silicone really present in the tailpipe, or in the sampling system? The Timko citations did not discuss tailpipe tubing.

We used the Timko citations in order to explain how the polydimethylsiloxane (SiO(CH3)2) contaminations are interfering to the AMS spectrum. We add a reference (Lamma, 2017) that shows the presence of such compounds in the car exhaust. (PhD Thesis (in French): Mise au point d'une méthode de mesure des siloxanes méthyliques volatils dans le biogaz et dans l'air ambiant et étude de leur impact sur les systèmes photocatalytiques, Dr Lina Lamaa, December 2017, Univ. Lyon, France.)

Line 215: "we conclude that both GDI5 and D4 emitted randomly oil droplets". Surely the emissions are not random, and surely the authors understand a little more than they are saying here. Do the authors mean that the lubrication oil emissions were not correlated with engine load, cycle period, etc.? Please improve the statement to represent a better scientific discussion.

What we want to state here is that the oil droplets were present in the exhaust emissions of GDI5, and D4, (or GDI3 and D3 with the new numeration) during the first minutes of the cycle but also later on the cycle. TEM grids are collected for few minutes, it is not an on-line technique able to identify the effect of acceleration/deceleration. In any case, we are not able to identify a correlation found between the oil droplet emissions

[Figure]

and the engine load or the speed or the cycle period. We modified the text as: "...we concluded that both GDI3 and D3 cars emitted some oil droplets (see also section 3.2 for TEM images), but we did not observe any evident correlation between the oil droplet emissions and the engine load or the speed or the cycle period."

Table 2: Define theta. R = cos(theta in degrees). The text suggests that theta is a different parameter to R, when it is not.

The angle theta is described in detail in Kostenidou et al. (2009) and it is an independent parameter of the correlation coefficient R. For the angle theta calculation, the two mass spectra are considered as vectors with x m/z dimensions (equations 2, 3 and 4 of Kostenidou et al. (2009)). The angle $\theta$ is a more sensitive metric for AMS mass spectra comparison that are often quite similar to each other. After the publication of Kostenidou et al. (2009) the angle theta has been used in many other publications (e.g., Kaltsonoudis et al., 2017 ACP, Florou et al., 2017 ACP, Paciga et al., 2016 ACP, etc.), for AMS mass spectra comparisons. We have already stated in the Table 2 caption the reference of Kostenidou et al. (2009) and that: "The angle $\theta$ provides a better comparison for small differences in the mass spectra (when R2 is less than 0.97)." However, we expanded this sentence: "...as this method treats the mass spectra as vectors. More details about this method are given in Kostenidou et al. (2009)."

Figure 4: OPAH is not readable in light yellow text.

We changed the color in dark yellow.

Figure 6: Add interpretation to the caption. I also suggest adding one of the micrographs from Figure S9 which show the tiny metallic particles, since this is a rare observation.

We transferred 3 micrographs from Figure S9 to Figure that 6 show the tiny metallic particles. The caption has been modified as follows: "Figure 6: TEM images of samples collected during Artemis urban cold cycle. The images show soot particles as fractals,

as agglomerated, details of primary soot particles and some oil droplets: (a-c) GDI1 vehicle (dilution ratio 40), the sample was collected during the first 120 sec of the cycle; (d-f) GDI3 vehicle (dilution ratio 46), the sample was collected during the first 120 sec of the cycle; (g-i) D1 vehicle (dilution ratio 40) the sample was collected during the first 300 sec of the cycle; D3 sampling the first 45 sec of the motorway cycle, dilution ratio 2, tiny metallic inclusion are observed in (l)".

Data availability: I recommend that the authors upload a table with the data of Figure7 instead of stating that the data are available on request.

The data of Figure 7 are provided in the supplement in Table S6 and it is already mentioned in the main text (Table S6 summarizes these EFs in $\mu$g km-1).

We updated this revised version adding the BC EFs of the PFI and the D3 vehicles as a co-author decided to give us these data after we submitted the manuscript in ACPD. We added some parts in the text where we discuss and compare these new EFs.

References:

[revised manuscript text omitted]